



# Vertical profiles of leaf photosynthesis and leaf traits, and soil nutrients in two tropical rainforests in French Guiana before and after a three-year nitrogen and phosphorus addition experiment

Lore T. Verryckt[1], Sara Vicca[1], Leandro Van Langenhove[1], Clément Stahl[2], Dolores Asensio[3,4], Ifigenia Urbina[3,4], Romà Ogaya[3,4], Joan Llusià[3,4], Oriol Grau[3,4,6], Guille Peguero[3,4], Albert Gargallo-Garriga[3,4,7], Elodie A. Courtois[8], Olga Margalef[3,5], Miguel Portillo-Estrada[1], Philippe Ciais[9], Michael Obersteiner[10], Lucia Fuchslueger[11], Laynara F. Lugli[12], Pere-Roc Fernandez-Garberí[3,4], Helena Vallicrosa[3,4], Melanie Verlinden[1], Christian Ranits[11], Pieter Vermeir[13], Sabrina Coste[14], Erik Verbruggen[1], Laëtitia Bréchet[1,15], Jordi Sardans[3,4], Jérôme Chave[16], Josep Peñuelas[3,4], Ivan A. Janssens[1]

[1] Plants and Ecosystems (PLECO), Biology Department, University of Antwerp, Wilrijk, 2610, Belgium
[2] UMR EcoFoG, AgroParisTech, CIRAD, CNRS, INRAE, Université des Antilles, Université de Guyane, Kourou, 97310, France.
[3] CREAF, Campus Universitat Autònoma de Barcelona, Cerdanyola del Vallès, Barcelona, 08193, Catalonia, Spain.
[4] CSIC, Global Ecology Unit CREAF- CSIC-UAB, Bellaterra, Barcelona, 08193, Catalonia, Spain.
[5] RISKNAT Research Group, Department of Earth and Ocean Dynamics, University of Barcelona, Barcelona, 08028, Spain
[6] Cirad, UMR EcoFoG (AgroParisTech, CNRS, Inra, Univ Antilles, Univ Guyane), Campus Agronomique, Kourou, 97310, French Guiana
[7] Global Change Research Institute, Czech Academy of Sciences, Bělidla 986/4a, Brno, 60300, Czech Republic.
[8] Laboratoire Ecologie, évolution, interactions des systèmes amazoniens (LEEISA), CNRS, IFREMER, Université de Guyane, Cayenne, French Guiana
[9] Laboratoire des Sciences du Climat et de l'Environnement, CEA-CNRS-UVSQ, Gif-sur-Yvette, France
[10] International Institute for Applied Systems Analysis (IIASA), Laxenburg, Austria
[11] Centre for Microbiology and Environmental Systems Science (CMESS), University of Vienna, Vienna, 1090, Austria
[12] Coordination of Environmental Dynamics, National Institute of Amazonian Research, Manaus, AM, 69060-062, Brazil
[13] Laboratory for Chemical Analysis (LCA), Department of Green Chemistry and Technology, Faculty of Bioscience Engineering, Ghent University, Ghent, 9000, Belgium
[14] UMR EcoFoG, AgroParisTech, CIRAD, CNRS, INRAE , Université des Antilles, Université de Guyane, Kourou, 97310, France
[15] INRA, UMR EcoFoG: Research Unit Ecology of Guianan Forests (CNRS, CIRAD, AgroParisTech, Université des Antilles, Université de Guyane), Kourou, 97310, France
[16] Evolution et Diversité Biologique, CNRS, IRD, UPS, 118 route de Narbonne, Toulouse, 31062, France

*Correspondence to*: Lore T. Verryckt (lore.verryckt@uantwerpen.be)





**Abstract.** Terrestrial biosphere models typically use the biochemical model of Farquhar, von Caemmerer and Berry (1980) to
simulate photosynthesis, which requires accurate values of photosynthetic capacity of different biomes. However, data on
tropical forests are sparse and highly variable due to the high species diversity, and it is still highly uncertain how these tropical
forests respond to nutrient limitation in terms of C uptake. Tropical forests often grow on phosphorus (P)-poor soils and are,
in general, assumed to be P- rather than nitrogen (N)-limited. However, the relevance of P as a control of photosynthetic
capacity is still debated. Here, we provide a comprehensive dataset of vertical profiles of photosynthetic capacity and important
leaf traits, including leaf N and P concentrations, from two three-year, large-scale nutrient addition experiments conducted in
two tropical rainforests in French Guiana. These data present a unique source of information to further improve model
representations of the roles of N, P, and other leaf nutrients, in photosynthesis in tropical forests. To further facilitate the use
of our data in syntheses and model studies, we provide an elaborate list of ancillary data, including important soil properties
and nutrients, along with the leaf data. As environmental drivers are key to improve our understanding of carbon (C)-nutrient
cycle interactions, this comprehensive dataset will aid to further enhance our understanding of how nutrient availability
interacts with C uptake in tropical forests. The data are available at DOI 10.5281/zenodo.4719242 (Verryckt, 2021).

## 1 Introduction

Tropical forests play a significant role in the global carbon (C) cycle, contributing more than one third of global terrestrial
gross primary productivity (GPP) (Beer et al., 2010; Malhi, 2010). To obtain accurate estimations of the global C budgets, a
thorough understanding of the functioning of these tropical forests is thus important. It is still highly uncertain how these
tropical forests, and in particular lowland tropical forests, respond to nutrient limitation and to global change in terms of C
uptake (Fleischer et al., 2019; Wieder et al., 2015).
Leaf photosynthetic capacity is the primary driver of C uptake and its accurate representation in terrestrial biosphere models
(TBMs) is essential for robust projections of C stocks and fluxes under global change scenarios. Photosynthesis in $C_3$ species
is typically represented in nearly every major large-scale TBM by the Farquhar, von Caemmerer and Berry (FvCB) model of
photosynthesis (Farquhar et al., 1980; von Caemmerer and Farquhar, 1981).
The FvCB model determines photosynthesis (*A*) by the most limiting of two processes, Rubisco activity and electron transport.
Empirical studies to determine the key parameters of these two processes (i.e. the maximum rate of carboxylation $V_{cmax}$, the
maximum electron transport rate, $J_{max}$) and to test empirically their limitations on the leaf- and canopy-scale are necessary for
obtaining the data required for parameterizing the FvCB model (Medlyn et al., 2015). Plant trait databases, now widely
available, offer an excellent opportunity for parameterizing models. However, data on tropical forests are sparse and highly
variable due to the huge species diversity (Rogers, 2014).
Moreover, empirical studies on leaf photosynthesis and leaf traits in tropical forests have mainly focused on upper canopy
leaves (i.e. Bahar et al. (2016), Berry and Goldsmith (2020), Rowland et al. (2015)) as a trade-off to cover a broader set of tree
species in these highly diverse tropical forests. Subsequently, light and leaf nitrogen (N) profiles are used to upscale leaf- to



canopy-level photosynthesis, as is also common practice in temperate and boreal forests (Bonan, 2015). However, the leaf N
gradient is shallower than the light gradient (Bonan, 2015) and accumulating evidence suggests a regulating role of phosphorus
(P) for photosynthesis in tropical trees growing on low-P soils (Walker et al., 2014; Mo et al., 2019; Norby et al., 2017; Bahar
et al., 2016). Consequently, vertical variation in leaf traits within the canopy of tropical forests is often not accounted for in a
proper way by TBMs.
Environmental drivers are key to improve our understanding of C-nutrient cycle interactions and our ability to model them.
Climate data are often directly available at high spatial resolution and at the global scale from databases such as Worldclim
(Ruiz-Benito et al., 2020), while observations of soil properties and soil nutrient availability are often missing (Vicca et al.,
2018). Soil variables have been shown to be strong predictors of leaf traits in higher plants (Maire et al., 2015). Comprehensive
soil data, including soil properties such as texture and pH as well as important nutrients, are needed to further enhance our
understanding of how and why nutrient availability interacts with C uptake in tropical ecosystems and their responses to global
environmental change (Vicca et al., 2018).
As most tropical forests are growing on highly weathered soils and contain N-fixing plants and free-living organisms, the
widely accepted ecological paradigm states that they tend to be limited by P rather than by N (Wright et al., 2018; Walker and
Syers, 1976). Nutrient addition experiments are a great asset to offset possible nutrient limitations and to see how the system
reacts (Vitousek and Howarth, 1991). Long-term nutrient addition experiments are important to study the role of leaf nutrients
in key role processes such as photosynthesis. However, in tropical forests only a few large-scale nutrient addition experiments
have been carried out and the results are ambiguous (Wright et al., 2018; Wright, 2019).
Here, we provide photosynthesis data and a set of leaf traits collected at multiple canopy levels at two forest sites in French
Guiana, as well as data on responses to two three-year, large-scale N and P nutrient addition experiments. Given the importance
of ancillary data such as environmental data and soil properties for model and synthesis studies (Vicca et al., 2018), we also
provide an extensive dataset of environmental data, including pre-treatment soil properties and nutrients.

## 88   2 Sampling sites

### 89   2.1 Study site description

The data were collected in French Guiana, South America at two old-growth, lowland tropical rainforest sites, Paracou and
Nouragues (Figure 1A, B). The climate in French Guyana is tropical wet, characterized by a wet and a dry season due to the
north-south movement of the Inter-Tropical Convergence Zone (ITCZ) (Bonal et al., 2008). From December to July, the ITCZ
brings heavy rains, which peak in May when monthly rainfall typically exceeds 600 mm. The dry season, with < 100 mm
rainfall each month, lasts from August to November, with an additional short, dry period in March. Mean annual air
temperature is near 26°C for both sites (Bongers et al., 2001; Gourlet-Fleury et al., 2004).
The first study site was situated at the Paracou Research Station (5°16′N, 52°54′W) (Figure 1B, D) and is characterized by an
average annual rainfall of 3100 mm year$^{-1}$ (2004 – 2015) (Aguilos et al., 2019). The average density of trees with a diameter



at breast height (DBH) > 10 cm is ca. 620 trees ha⁻¹, and tree species richness averaged ca. 160 species ha⁻¹ (Bonal et al., 2008).
The mean canopy height was 35 m, with emergent trees exceeding 40 m. The second study site, the Nouragues Research
Station (4°02′N, 52°41′W) (Figure 1B, C), is located 120 km south of Cayenne. Here we sampled at two locations, near the
Inselberg station (Petit Plateau) and near the Pararé station (COPAS). The Nouragues forest receives approximately
3000 mm rain year⁻¹ and tree density (DBH > 10 cm) averaged ca. 535 trees ha⁻¹ (Bongers et al., 2001). Tree species richness
ranges between 180 and 200 species ha⁻¹. The canopy height varies between 30 and 40 m, with emergent trees reaching 60 m
(Van Der Meer et al., 1998).
The soils at Paracou are derived from the Bonidoro series, characterized by schist and sandstones and locally crossed by veins
of pegmatite, aplite and quarsites (Gourlet-Fleury et al., 2004; Epron et al., 2006), whereas the soils of Nouragues have a
weathered granite parent material of the Caraibe series (van der Meer and Bongers, 1996; Bongers et al., 2001). According to
the USDA texture classification chart, the soils at Paracou range from loamy sand to sandy loam and at Nouragues from sandy
loam to silty clay (Van Langenhove et al., 2019). These old and highly weathered soils at both sites are characterized as
nutrient-poor Acrisols (FAO, 1998) and  are, compared to the generally younger, nutrient-richer soils of western Amazonia,
particularly low in P concentration (Hammond, 2005; Grau et al., 2017).
At both forest sites, an instrumentation tower is in place at which meteorological measurements and measurements of
ecosystem net $CO_2$ exchange with the eddy covariance technique have been carried out on a continuous basis since 2003 in
Paracou (Bonal et al., 2008) and since 2014 at Nouragues.

**2.2 Nutrient addition experiment**

**2.2.1 Set-up**

In 2015, we set up twelve 50 x 50 m plots, in three blocks of four plots, along a toposequence at Paracou and Nouragues,
resulting in a total of 24 plots. These blocks were located at distinct landscapes: (1) bottom, i.e. just above the creek running
through the valley, (2) slope, i.e. the intermediate section of the elevation and (3) top, i.e. where the slope evens out and
becomes the hilltop (Courtois et al., 2018). The valley bottoms and hilltops differ ca. 20–50 m in elevation over horizontal
distances of 200–400 m (Van Langenhove et al., 2019), with maximum altitudes of ca. 70 m and 120 m for Paracou and
Nouragues, respectively (Courtois et al., 2018).
In October 2016, a field nutrient addition experiment at both sites was initiated and is ongoing to this day. In each block, one
plot served as control plot and the remaining three plots received one of three nutrient addition treatments (+N, +P or +NP).
Fertilizer was applied twice per year by hand-broadcasting commercial urea ($(NH_2)_2CO$) and/or triple superphosphate
($Ca(H_2PO_4)_2$) at a rate of 125 kg N ha⁻¹ y⁻¹ (+N treatment) or 50 kg P ha⁻¹ y⁻¹ (+P treatment), or both amounts together (+NP
treatment). These application rates are identical as those in the ongoing nutrient addition experiment in Barro Colorado Nature
Monument in Panama, which was initiated in 1998 (Wright et al., 2011), and the Amazon Fertilization Experiment (AFEX)
near Manaus in Brazil, initiated in 2017 (Lugli et al., 2021), to enable future comparison of our results. In Paracou, these rates





of nutrient addition represent, respectively, 130% and 250% of yearly N and P input through litterfall and atmospheric
deposition combined (Van Langenhove et al., 2020b).
Leaf and soil sampling took place before (in 2015) and after a three-year period (in 2019) of the nutrient addition experiment.
To avoid border effects of the nutrient addition, these measurements were conducted in the central 20 x 20 m area within the
larger 50 x 50 m plots.

### 2.2.2 Fertilizer composition

Within the nutrient addition experiment, N was added as commercial urea ($(NH_2)_2CO$) and P as triple superphosphate
($Ca(H_2PO_4)_2$). The chemical composition of the applied fertilizers was analyzed to know the exact composition. Samples of
both fertilizers were dried at 70°C for 48 h, after which they were ground. Total N of the fertilizers was determined by dry
combustion using a Skalar Primacs (Skalar Holding, The Netherlands). $P_2O_5$ and MgO in mineral acid where determined by
an iCAP 7400 radial optical emission spectrometer (Thermo Fisher Scientific, Germany). The ground samples were analyzed
with an iCAP 7400 radial optical emission spectrometer (Thermo Fisher Scientific, Germany) to determine the potassium (K),
calcium (Ca) and magnesium (Mg) concentrations, as well as the heavy metal concentrations (arsenic (As), cadmium (Cd),
chromium (Cr), copper (Cu), iron (Fe), nickel (Ni), lead (Pb), zinc (Zn), molybdenum (Mo)).

### 3 Data and Methods: soil sampling

### 3.1 Sampling design

In 2015, we sampled soil to a depth of 30 cm, according to a five-on-dice sampling pattern within the 20 x 20 m plots (Figure
2). At each sampling point, we sampled bulk density at a depth of 0-15 cm and 15-30 cm using an auger with a 15-cm long
cylindrical head (8-cm diameter). Additionally, we took three soil cores with a gouge auger (30-cm length, 5-cm diameter).
These three cores were split into two depths (0-15 cm and 15-30 cm), pooled together per depth and used for gravimetric soil
water content determination, soil particle size distribution analysis and chemical analysis after sieving (< 2 mm). We divided
the soil into a 'surface' layer (0–15 cm) and a 'deeper' layer (15–30 cm) instead of sampling by generic horizon because the
upper horizon at both sites varies between 0–15 and 0–20 cm depth while the next horizon often extends to 50 cm depth and
beyond (Van Langenhove et al., 2020c; Guitet et al., 2016; Bongers et al., 2001). In 2019, after 3 years of nutrient addition,
the soil sampling was repeated. However, these data have not been processed yet and will be made available through
publication as soon as possible.

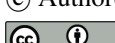



### 3.2 Soil physical properties

### 3.2.1 Bulk density

We sampled soil bulk density in the wet season of 2015. In each plot, we took five cores at two depths and these samples were sieved through a 2-mm sieve. We collected the soil fraction, the roots and the stones, which were dried and weighed separately at 105°C for 24 h. In our database, we report two measures of bulk density: inclusive bulk density is the weight of the dried soil core divided by its volume (i.e. the volume of the auger, 754 cm³), whereas exclusive bulk density is calculated by dividing the total weight of the soil fraction (excluding roots and stones) by the volume of the entire core (i.e. 754 cm³).

### 3.2.1 Soil particle size distribution

The particle size distribution at plot level was analysed only in the wet season of 2015, assuming it would not change with seasonality. Therefore, we mixed by hand the five samples per plot that were sieved (< 2 mm) after extraction using a gouge auger, and analyzed these mixed samples as one composite sample per depth and per plot. We determined the soil particle size distribution using sedimentation with the hydrometer method (Gee and Bauder, 1986) after SOM oxidation with $H_2O_2$, as described in protocol 1.3.5 Soil texture in the Supporting information S1 Site characteristics and data management in Halbritter et al. (2020). Soil particles were dispersed with sodium hexametaphosphate and the quantity of sand, silt and clay were determined using a hydrometer.

### 3.2.3 Soil moisture

The gravimetric soil water content (%) was determined in both the wet and the dry season of 2015. We weighed roughly 10 g of fresh soil, which was then dried at 70°C to constant mass and weighed to obtain the dry mass. The gravimetric water content is calculated as the mass of water (i.e. the difference in mass weight of fresh and dried soil) per mass of dry soil.

### 3.3 Soil nutrients

### 3.3.1 Chemical analyses: concentrations and availability

Freshly sieved soil was used for the measurement of pH and the extraction of inorganic N ($N_i$) and inorganic P ($P_i$). We measured the soil pH using a pH meter (HI 2210-01, Hanna Instruments, USA) after adding 1M KCl to the soil in a 1:2.5 w:v ratio and shaking it for 1 h. The same solution was passed through a 42-µm filter and the filtrate's concentration of $NH_4^+$ and $NO_3^-$ was determined colourimetrically (SAN++ continuous flow analyzer, Skalar Inc., The Netherlands). The $P_i$ was extracted with the Olsen-P bicarbonate extraction (Olsen et al., 1954) and measured on an iCAP 6300 Duo ICP optical emission spectrometer (Thermo Fisher Scientific, Germany).

Sieved soil samples were dried at 60°C to constant mass and was then ground in a ZM 200 ball mill (Retsch GmbH, Haan, Germany). We extracted $P_i$ on previously dried soil with the Bray P acid fluoride extraction (Bray and Kurtz, 1945) followed



by analysis on an iCAP 6300 Duo ICP optical emission spectrometer (Thermo Fisher Scientific, Germany). Water soluble
molybdate ($MoO_4^{2-}$) and phosphate ($PO_4^{3-}$) were determined through resin extraction on previously dried (60°C) soil
(Wurzburger et al., 2012). The soil samples were mixed with water in a 1:6 ratio and five 2-cm² strips of anion-exchange
membrane (VWR Chemicals, USA) were added (Van Langenhove et al., 2019; Wurzburger et al., 2012). After stirring this
mixture 24 h, the strips were rinsed and eluted with 10% $HNO_3$. The concentrations of water soluble $MoO_4^{2-}$ and $PO_4^{3-}$ were
analyzed with an iCAP 6300 Duo ICP optical emission spectrometer (Thermo Fisher Scientific, Germany).
Additionally, we analyzed total macro- and micronutrient concentrations on previously dried soil. Soil C and N concentrations
were determined by dry combustion with a Flash 2000 elemental analyzer (Thermo Fisher Scientific, Germany), and total soil
C and N was analyzed by EA-IRMS (EA1110, CE Instruments, Milan, Italy), coupled to a Finnigan MAT Delta Plus IRMS
(Thermo Fisher Scientific, Germany). The concentrations of P, K, Ca, Mg, sulphur (S), manganese (Mn), sodium (Na),
vanadium (V), strontium (Sr), As, Cd, Cr, Cu, Fe, Ni, Pb, Zn and Mo were obtained by acid digestion in an ultraWAVE
digestor (Milestone, Italy), followed by ICP-MS (7500ce model, Agilent Technologies, Tokyo, Japan) analysis.
**3.3.2 PRS probes**
We used Plant Root Simulator (PRS ®) probes (Western Ag Enterprises, Inc., Saskatoon, CA) to provide proxies for plant
available ions in soil solution (Van Langenhove et al., 2020a; Halbritter et al., 2020). These PRS probes are ion exchange resin
membranes (IEM; approximately 5.5 cm × 1.6 cm, or 17.5 cm² including both sides of the IEM) held in plastic supports that
were vertically inserted into soil with minimal disturbance (Hangs et al., 2004). In each plot, we installed five root exclusion
cylinders (REC) following a five-on-a-dice design to study soil nutrient dynamics in absence of the root system (Van
Langenhove et al., 2020a). RECs were PVC collars inserted 20 cm into the soil to sever all near-surface roots and mycorrhizal
fungal hyphae, and were installed prior to the first soil nutrient measurements in May 2015.
In both the wet (May) and dry (October) season of 2015, we installed four anion and four cation probes, which formed one
sample, into each REC. After a period of two weeks, the PRS™ probes were removed from the soil, washed with distilled
water and shipped to the manufacturer for analysis of the following nutrients: P, K, Ca, Mg, S, Mn, Cd, Cu, Fe, Pb, Zn, nitrate
($NO_3$), ammonium ($NH_4$), boron (B) and aluminium (Al). In situ burial of the PRS™ probes provide a dynamic measure of
nutrient flux, or nutrient supply rate to an ion sink (Gibson et al., 1985; Casals et al., 1995). The results are thus expressed as
the amount of nutrient adsorbed per surface area of IEM during the duration of burial (Qian and Schoenau, 2002). In our study
it was expressed as μg nutrient 10 cm$^{-2}$· for two weeks.
**4 Data and Methods: leaf sampling**
**4.1 Tree selection**
In each 20 x 20 m plot at Paracou and Nouragues, we selected five mature trees, of which three were top-canopy species likely
dominating the functioning of the plot and two trees were sub-canopy species. Additionally, we selected twelve trees at the



COPAS site near the Pararé station at Nouragues. At Paracou we selected in each 50 x 50 m plot four species of saplings,
defined as a tree < 2 m in height, with at least ten individuals per plot. This resulted in seven sapling species selected in total
and 40 individuals selected per plot. Overall, we selected 131 trees belonging to 76 different tree species differing in relative
abundance (Figure 3, Table 2).
**4.2 Photosynthesis**
**4.2.1 Leaf gas exchange**
We measured leaf gas exchange measurements using a set of infrared gas analyzers (IRGAs) incorporated into a portable
photosynthesis system (LI-6400XT, LI-COR, Lincoln, NE, USA). A leaf was clamped within a chamber with controlled
microenvironmental conditions during the measurements and the device measures the concentration changes of water vapour
($H_2O$) and carbon dioxide ($CO_2$) between incoming and outgoing air. The relative humidity inside the chamber (67.9 % ± 0.06)
was kept as close to ambient as possible during the measurements and the air flow rate was set at 500 µmol s$^{-1}$. The chamber
block temperature was controlled to minimize variation in leaf temperature and was set at 30.1 ± 0.9°C.
Pre-treatment measurements of the mature trees were carried out in both wet (May-June) and dry season (October-November)
of 2015, except for the wet season measurements of the plots at Paracou situated at the slope which were measured in June
2016 due to practical constraints. After three years of nutrient addition, we repeated the leaf gas exchange measurements of
the mature trees in the wet season (May-June) of 2019. We measured leaf gas exchange of leaves collected at two different
canopy heights, estimated relative to the top of the canopy: sunlit, upper canopy foliage and shaded, lower canopy foliage.
Branch excision prior to measuring leaf gas exchange was necessary to reach the canopy leaves. The excised branches (ca.
2 m-long) were cut by a tree climber and immediately recut under water to restore hydraulic conductivity (Domingues et al.,
2010; Dusenge et al., 2015; Rowland et al., 2015; Verryckt et al., 2020b). Photosynthesis measurements of the saplings were
carried out on leaves still attached to the trees and were performed in August 2016 (pre-treatment) and August 2017 (1-year
of nutrient addition). All photosynthesis measurements were conducted between 09:00 and 16:00 (local time).
**4.2.1 Photosynthetic $CO_2$-response curves**
Photosynthetic $CO_2$-response curves (Figure 4) were established by measuring net photosynthetic rates ($A_n$) at different $CO_2$
concentrations by controlling the reference $CO_2$ concentrations, while maintaining a constant temperature and photosynthetic
photon flux density (PPFD). The $A_n$-$C_i$ ($C_i$, the $CO_2$ concentration of the leaf intercellular spaces) measurements began at the
ambient $CO_2$ concentration of 400 ppm. Once a steady state of photosynthesis was reached, the $CO_2$ concentrations was
reduced stepwise to 50 ppm, then returned to 400 ppm, and thereafter increased to 2000 ppm, to obtain a total of 10-14
measurements per leaf. We measured $A_n$-$C_i$ curves for one to three leaves per canopy level, resulting in a total of 1708 curves
measured (Table 1). The methods used here are described in protocol 2.1.3 Leaf-scale photosynthesis in the Supporting



Information S2 Carbon and nutrient cycling in Halbritter et al. (2020). Additionally, we measured leaf dark respiration ($R_d$) by
taking five consecutive measurements on one leaf per branch level, after keeping the branch in complete darkness for 30 min.
The $A_n$-$C_i$ curves were carried out at a saturating PPFD level of 500 and 1300 µmol m$^{-2}$ s$^{-1}$ for shaded, lower canopy and sunlit,
upper canopy leaves, respectively. Verryckt et al. (2020a) showed that 500 µmol m$^{-2}$ s$^{-1}$ is below light-saturation and a PPFD
level of 800 µmol m$^{-2}$ s$^{-1}$ would have been optimal for shaded, lower canopy leaves. Increasing the PPFD level would, however,
only result in a small increase of net photosynthesis.
The $A_n$-$C_i$ curves present $A_n$ at any given $C_i$ as the minimum of three potential limitations: rubisco ($V_{cmax}$), ribulose 1,5-
biphosphate ($J_{max}$) or triose phosphate use (TPU). These curves were fitted with the FvCB model using the fitaci function from
the "Plantecophys" package (Duursma, 2015) in R 3.3.3 (R Core Team, 2019) to obtain the biochemical parameters $V_{cmax}$, $J_{max}$
and, when possible, TPU.

**4.2.2 Light-saturated photosynthesis ($A_{sat}$) of saplings**

Light-saturated photosynthesis ($A_{sat}$) and the stomatal conductance ($g_s$) were measured, separately from the $A_n$-$C_i$ curves, for
the saplings at the Paracou plots situated at the bottom valleys and hilltops. We used the methods described in protocol 2.1.3
Leaf-scale photosynthesis and protocol 5.7 Stomatal conductance in the Supporting Information, respectively, S2 Carbon and
nutrient cycling and S5 Stress physiology in Halbritter et al. (2020). The $CO_2$ concentration was maintained at 400 ppm and a
the PPFD was set at 500 µmol m$^{-2}$ s$^{-1}$. In 2016, $A_{sat}$ and $g_s$ were measured on one leaf per individual and for three to five
individuals per species. This resulted in a total of 159 measurements. In 2017, after 1-year of nutrient addition, these
measurements were repeated on the same individuals when they survived, and on additional saplings resulting in 195
measurements in total.

**4.3 Leaf traits**

All leaves used for gas exchange were collected and, for the mature trees, leaf chlorophyll was estimated by averaging five
chlorophyll content measurements from each leaf with a CCM-200 portable chlorophyll meter (Opti-Sciences, Tyngsborough,
MA, USA), thereby avoiding major veins and areas of obvious visual damage or disease. A unitless chlorophyll content index
(CCI) value was calculated from the ratio of optical absorbance at 655 nm to that at 940 nm. Then, the leaves were scanned to
obtain the leaf area, dried for at least 48 h and weighed. Leaf area was measured with the leaf area meter LI-3100C (LI-COR,
Lincoln, NE, USA), which, together with the leaf dry mass, was used to calculate the specific leaf area (SLA). After grinding
the dried leaves, leaf nutrient concentrations were determined in accordance with protocol 2.1.6 Foliar nutrient stoichiometry
and resorption in Supporting information S2 Carbon and nutrient cycling in Halbritter et al. (2020). Leaf C and N
concentrations of 10 mg subsamples were determined by dry combustion using a Flash 2000 elemental analyzer (Thermo
Fisher Scientific, Germany). Leaf P concentration of 0.1 g subsamples was determined by an acid digestion method (Walinga
et al., 1989) on a Skalar SAN++ continuous flow analyzer, and thereafter analyzed on an iCAP 7400 radial optical emission





spectrometer (Thermo Fisher Scientific, Germany) to determine the concentration of leaf K, Ca, Mg, and Mn. Samples were
ashed, digested with nitric acid ($HNO_3$) and filtered prior to analysis.
In the wet season of 2015, we also analyzed leaf S, Na, V, Sr, As, Cd, Cr, Cu, Fe, Pb, Zn and Mo on a model 7500ce ICP-MS
spectrometer (Agilent Technologies, Tokyo, Japan) after digestion with $HNO_3$.
For saplings, total phenolic concentration of the leaves was measured using an improved Folin-Ciolcalteu assay (Singleton
and Rossi, 1965; Marigo, 1973) and total leaf tannin concentration of the leaves was determined with the butanol/HCl method
(Porter et al., 1985) modified as in Makkar and Goodchild (1996). The extracts for both phenol and tannin concentrations were
determined using a Helios Alpha spectrophotometer (Thermo Spectronic, Cambridge, UK) at 760 and 550 nm, respectively.
Both methods are described in detail in Peñuelas et al. (2010).
**4.4 Ancillary data**
The vertical structure of a tropical rainforest is complex and multi-layered, resulting in great variation in light availability
within the canopy (Yoda, 1974). We assessed the light environment of each studied tree by visually estimating the canopy
light exposure or Dawkins' crown illumination index (Dawkins, 1958). This index describes a tree's light environment based
on a five-point scale ranging from (1) no direct light for suppressed trees to (5) crown fully illuminated for emergent trees
(Figure 5).
Sampling height of the mature trees was measured with a Forestry Pro rangefinder (Nikon, Tokyo, Japan) by tree climbers
situated at sampling height pointing towards the soil. For saplings, we measured the total height of the tree using a measuring
tape, and additionally we measured the diameter at 10 cm and at 50 cm above surface level. On top of each sapling, we
measured leaf area index (LAI) with the LAI-2000 (LI-COR, Lincoln, NE, USA) during periods of overcast sky.
Herbivory rates, i.e. foliar damage by herbivores, of the saplings was estimated as punctual herbivory (%) according to Pirk
and Farji-Brener (2012). We visually assessed the missing area of the leaf and assigned each leaf to the following categories:
0, 0.1-5, 5.1-25, 25.1-50, 50.1-100 % area consumed. We calculated the percentage of foliar damage per sapling by multiplying
the number of leaves of each category by the mid-point foliar damage of each category (i.e. 0, 2.5, 15, 37.5, 75 % respectively)
and dividing this result by the total number of leaves per sapling. To test the accuracy of this method, we photographed 106
leaves we visually assessed and compared visual estimations of herbivory in the field using ImageJ. In 90% of the cases
categories were well assigned.
**5 Challenges of fieldwork in the tropics**
Fieldwork in the tropical rainforest is challenging because of its remoteness and extremely moist and warm climate. While
Paracou is only a 45 min-drive from a major city, Kourou, Nouragues can only be reached by helicopter (30 min from Cayenne,
the capital of French Guiana) or a combination of car (2 h from Cayenne) and motorized canoe (4-6 h from Regina to
Nouragues). Carrying out fieldwork in such a remote location requires adequate logistical planning and funding, as both people





and material need to be transported to this remote site and stay there for prolonged periods. Proper planning and coordination
with the whole team was required to get the fieldwork finished within a limited timeframe.
The combination of high temperature and humidity poses an additional hurdle as this generally makes physical exertion harder
than in temperate climates and, most importantly, decreases the longevity of most if not all electronic devices. Indeed, we
suffered from several Li-6400XT malfunctions, as well as defects of laptops, freezers and drying ovens. However, these defects
did not reduce the reliability of our data, but it required extra precautions and increased expenses. We tested, for example, the
Li-6400XT devices each morning and the devices were regularly cross-calibrated. Malfunctions of these devices led to
troubleshooting and extra testing before new measurements were carried out, which can be very time consuming and did have
an impact on the amount of data that could be gathered. Laptops and other electronic devices were best stored overnight in a
waterproof bag/barrel, whereas regularly moving them in and out of air-conditioned rooms increased malfunctions.
Additionally, access to power was limited and a portable generator was often required to carry out all photosynthesis
measurements.
The tropical soil is hard, making it very labour intensive to take soil cores and posing several other problems. Soil corers
deformed as they are not developed for tropical soil types and the installation of PRS probes without breaking them was very
challenging. Another challenge is reaching upper canopy leaves up to > 50 m height above ground level, which required
technical tree climbing skills and equipment from experienced tree climbers.
High species diversity and stand structural complexity of tropical forests are a major challenge to understand the ecosystem
functioning of tropical forests and force researchers to study either some abundant species following them in time or to take
into account the high diversity limiting the number of replicates per species.
**6 Data availability**
This Photosynthesis-Soil database is provided as an excel workbook and is freely available at DOI 10.5281/zenodo.4719242.
Photosynthetic $CO_2$-response curves are presented as raw $A_n$-$C_i$ files which can be analyzed by any user of the database, but
$V_{cmax}$ and $J_{max}$ values, fitted using the "Plantecophys" package (Duursma, 2015) in R, can also be found in the database in
addition to the leaf traits. The soil database of 2015, including bulk density, soil particle size distribution, soil moisture and
nutrients, and the fertilizer composition are shown in separate sheets.
**7 Summary**
Publicly accessible and usable datasets from experimental sites are needed to greatly enhance the power of data synthesis as
well as model development and evaluation (Vicca et al., 2018; Halbritter et al., 2020). We provide vertical profiles of
photosynthetic capacity data and important leaf traits from two three-year large-scale nutrient addition experiments conducted
in two tropical rainforests in French Guiana. Our dataset is extremely valuable to the modelling and tropical ecology



community as we present a valuable source of information to further improve model representations of the roles of leaf
nutrients in photosynthesis in tropical forests. We present raw $A_n$-$C_i$ curves, which allows the curves to be fit under the same
assumptions as curves collected at other sites, avoiding bias in the method of analysis. In addition, we provide $V_{cmax}$ and $J_{max}$
values making these values immediately available to the modelling community. We provide leaf-level photosynthesis data at
several heights within the canopy from mature trees and saplings allowing to study differences in sunlit and shaded leaves.
Ancillary data such as herbivory and leaf phenol concentration can be of great value as additional data to other studies on these
topics. A large set of soil properties and nutrient availabilities in the soils underlying the studied trees were made available as
these data are highly relevant to understand how and why nutrient availability interacts with C uptake in tropical forests.

## Author contribution

L.T.V., L.V.L., C.S., D.A., I.U., R.O., J.L., O.G., G.P., A.G.G., E.A.C., O.M., P.C., M.O., L.F., L.L, , P.R.F.G., H.V., M.V.,
C.R., and I.A.J. contributed to the field work and collected the data. P.R.F.G., M.P.E. and P.V. carried out the majority of the
lab analysis. The manuscript was drafted by L.T.V., I.A.J. and S.V. and was further revised by all co-authors.

## Competing interests

The authors declare that they have no conflict of interest.

## Acknowledgements

L. T. Verryckt is funded by a PhD fellowship from the Research Foundation Flanders (FWO). This project was funded by the
European Research Council Synergy Grant; ERC-2013-SyG-610028 IMBALANCE-P. We thank the staff of the Nouragues
Ecological Research Station, managed by USR mixte LEEISA (CNRS; Cayenne), and the Paracou station, managed by UMR
Ecofog (CIRAD, INRA; Kourou). Both research stations received support from "Investissement d'Avenir" grants managed by
Agence Nationale de la Recherche (CEBA: ANR-10-LABX-25-01, AnaEE-France: ANR-11-INBS-0001).
We are grateful to Valentine Alt, Samuel Counil, Jocelyn Cazal, Gonzalo Carrillo, Jean-Loup Touchard, Anthony Percevaux,
Benjamin Leudet and Stefan van Beveren for climbing the trees to collect the selected branches.

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





# Tables

**Table 1 Overview of the number of $A_n$-$C_i$ curves measured in this study. Measurements were carried out in the wet and dry season, at different canopy levels (U = upper canopy, sunlit leaves; L = lower canopy, shaded leaves; M = middle canopy leaves), at different field sites (Nouragues-Inselberg, Nouragues-Pararé, Paracou). The measurements carried out on saplings instead of on mature, canopy trees are mentioned separately. Pre-treatment measured were carried out in 2015 and 2016, whereas post-treatment (after 1 year and 3 years of nutrient addition) were carried out in 2017 and 2019.**

| | Nouragues-Inselberg | | | | Nouragues-Pararé | | | Paracou | | | | Paracou-Saplings |
|---|---|---|---|---|---|---|---|---|---|---|---|---|
| | U | | L | | U | M | L | U | | L | | |
| | wet | dry | wet | dry | dry | dry | dry | wet | dry | wet | dry | |
| **2015** | 141 | 82 | 146 | 76 | | | | 61 | 113 | 56 | 120 | |
| **2016** | | | | | | | | 42 | | 35 | | 38 |
| **2017** | | | | | 37 | 45 | 32 | | | | | 46 |
| **2019** | 156 | | 163 | | | | | 155 | | 164 | | |





**Table 2 Overview of the amount of mature trees and species measured in each plot at Nouragues-Inselberg (NOU-I), Nouragues-**
**Pararé (NOU-P) and Paracou (PAR). Values between brackets indicate the amount of different species measured per plot, in case**
**this differed from the amount of trees measured per plot. Names of plots are marked by a letter describing the topography (B =**
**bottom, S = slope, T = top) and a number describing the nutrient addition treatment (1 = +N, 2 = +NP, 3 = +P, 4 = control), in**
**agreement with Figure 1C, D.**

| | | 2015 | | 2016 | 2017 | 2019 |
|---|---|---|---|---|---|---|
| | | DRY | WET | WET | DRY | WET |
| **NOU-I** | B1 | 3 | 5 | | | 5 |
| | B2 | 2 | 3 | | | 2 |
| | B3 | 4 | 3 | | | 4 |
| | B4 | 5 | 5 | | | 5 |
| | S1 | 4 | 3 | | | 4 (3) |
| | S2 | 3 | 5 (4) | | | 5 (4) |
| | S3 | 4 | 5 (4) | | | 5 (4) |
| | S4 | 3 | 5 | | | 5 |
| | T1 | 4 | 5 | | | 5 |
| | T2 | 4 | 5 | | | 5 |
| | T3 | 3 | 5 | | | 5 |
| | T4 | 3 | 5 | | | 5 |
| **NOU-P** | COPAS | | | | 12 (9) | |
| **PAR** | B1 | 5 | 5 | | | 5 |
| | B2 | 5 | 5 | | | 5 |
| | B3 | 5 | 5 | | | 4 |
| | B4 | 5 (4) | 5 (4) | | | 5 (4) |
| | S1 | 5 | | 5 | | 5 |
| | S2 | 5 (4) | | 4 (3) | | 5 (4) |
| | S3 | 4 | | 4 | | 5 |
| | S4 | 4 | | 3 | | 5 (4) |
| | T1 | 4 | 5 (4) | | | 5 (4) |
| | T2 | 5 (3) | 4 (3) | | | 4 (3) |
| | T3 | 5 | 4 | | | 5 |
| | T4 | 4 | 4 | | | 5 |






## Figures

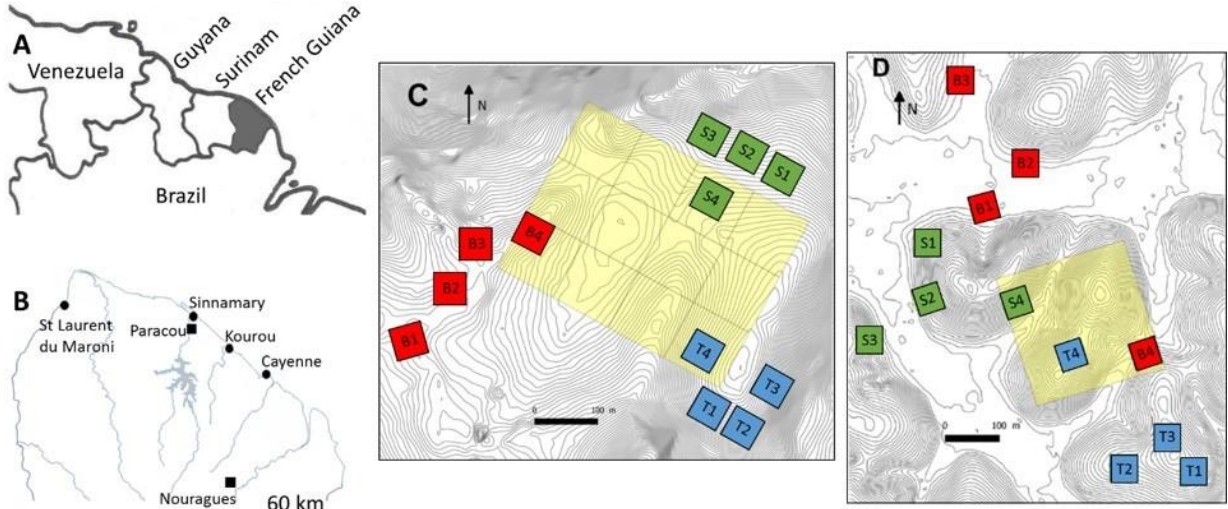

**Figure 1 Situational map: (A) north-eastern part of South America, (B) northern French Guiana with its main cities (circles) and the experimental sites Paracou and Nouragues (closed squares), (C) the twelve study 50 x 50 m plots of this study at Nouragues-Inselberg, (D) the twelve study 50 x 50 m plots of this study at Paracou. Adapted from Ferry et al. (2010) and Courtois et al. (2018). Plots are marked by a letter describing the topography (B = bottom, S = slope, T = top) and a number describing the nutrient addition treatment (1 = +N, 2 = +NP, 3 = +P, 4 = control).**

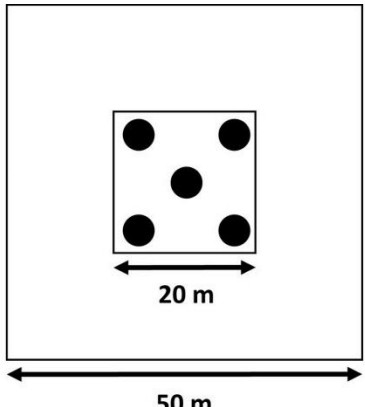

**Figure 2 Soil sampling design according to a 5-on-dice pattern, performed in the central 20 x 20 m area within each of the 50 x 50 m plots.**




Figure 3 Number of mature trees we sampled per species (bars) and the relative abundance for each species (dots) are shown for (A) Paracou and (B) Nouragues. Relative abundance was calculated as the percent composition of each species relative to the total number of that species with all 50 x 50 m plots at Paracou or Nouragues.

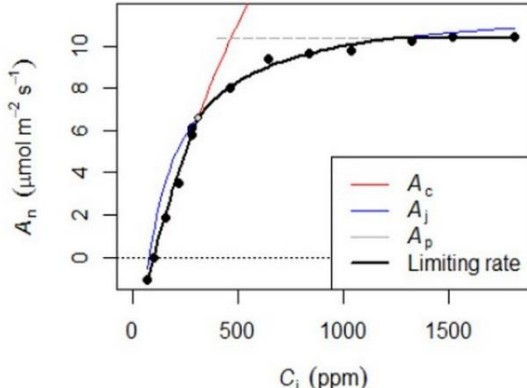

551

**Figure 4 Example of the $A_n$-$C_i$ response. Net photosynthesis ($A_n$) at any given $C_i$ is the minimum of three potential limitations:**
**rubisco ($A_c$), RuBp ($A_j$) or TPU ($A_p$).**

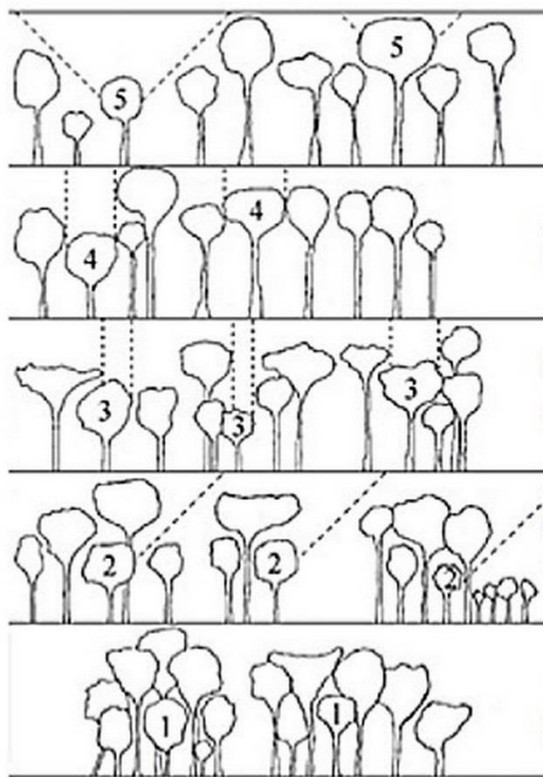

554

**Figure 5 Classification of the Dawkins' crown illumination index (adapted from Dawkins 1958): (1) no direct light, (2) low lateral**
**light, (3) some vertical light (10-90% of the vertical projection of the crown exposed to vertical illumination), (4) crown completely**
**exposed to vertical light, but lateral light blocked within some or all of the 90° inverted cone encompassing the crown (5) crown fully**
**exposed to vertical and lateral illumination.**