# Peer review of "Vertical profiles of leaf photosynthesis and leaf traits, and soil nutrients in two tropical rainforests in French Guiana before and after a three-year nitrogen and phosphorus addition experiment"

_Earth System Science Data, 2021_

## Author Response (AR1)

Dear editor,

We are pleased to resubmit our manuscript entitled "Vertical profiles of leaf photosynthesis and leaf traits, and soil nutrients in two tropical rainforests in French Guiana before and after a three-year nitrogen and phosphorus addition experiment" for consideration by Earth System Science Data.

We thank the subject editor and the reviewers for their constructive criticisms and we hereby submit our revised manuscript.

We have addressed each of the concerns as outlined below. Each comment is written in black text. Our responses are written in blue text and the changes made to the manuscript are written in italics and in between quotation marks, unless otherwise specified.

Sincerely,

Lore Verryckt, on behalf of the co-authors.

**Comments raised by Reviewer 1:**

This is an excellent and (almost) complete dataset. However, it could be even more valuable if some data on monthly rainfall during the experiment were also available.

We thank the reviewer for this positive assessment of our work. We have requested and received the monthly rainfall data at both sites during the period of the experiment (2015-2019) and included these data in the dataset. In section 2.1 Study site description we added the following:

*''The monthly rainfall data measured at the instrumentation towers of Paracou and Nouragues are shown in the dataset.''*

In addition, you should use the insolation index instead of the T/S/B symbol making the description a quantitative one. Fig. 1 suggests that you have the contours representing the topography of the sites, which is needed to calculate the insolation index (see, e.g., ArcMap by ESRI for a tool to calculate the insolation).

The B/S/T symbol indicates the topography of our plots, referring to the plots situated at the bottom of a valley, the slope or intermediate section of the elevation, and hilltops where the slope evens out. These valley bottoms and hilltops only differ maximal 20-50m in elevation over horizontal distances of 200-400m. The differences in topography result in differences in soil structure (more sandy at the bottom and more clayey at the top) and thus nutrient availability. All plots are situated in lowland tropical forests with high incoming solar radiation. It follows that the solar energy that is incident on these plots over a set period of time is similar. Therefore, we do not see the added value of the insolation index and we did not include any changes concerning this comment to our revised manuscript.

**Comments raised by Reviewer 2:**

This study presents a complete dataset for leaf and soil nutrient traits in two rainforests under treatments of nitrogen and phosphorus addition. This dataset can help improve our understanding of the changes of leaf and soil traits under nutrient additions in the P-limited forest ecosystems, and help improve model representations of forest carbon and nutrient cycles through providing parameters and mechanisms. The manuscript is well written. The experimental methods and dataset are described in detail.

We thank the reviewer for this positive assessment of our work and the constructive suggestions that helped to further improve the manuscript.

However, there are some serious deficiencies in this manuscript as described as below:

- The manuscript only presents the experiments, dataset and descriptions, but there is lack of a detailed analysis on the results (Section: Results and Analysis). Without the analysis results, the readers did not really know how the impacts of nutrient addition on leaf and nutrient traits.

We do not agree with this comment, since this manuscript is a data description paper and, therefore, detailed analysis of the data are outside the scope as described in the requirements for different manuscript types on the website of ESSD.

- A follow-up problem is that the measured data are not showed in figures or tables in the manuscript. Some key data should put in the manuscript rather than only in the excel sheets in the data website (zenodo).

We agree with the referee and provide new figures showing the main results of the photosynthesis and soil data.

We added table 1 showing the total and available nutrient concentrations of the surface soil (0-15cm) measured in 2015 in the three landscape positions (bottom, slope and top), separately for the wet and the dry season.

**Table 1 Surface soil (0-15 cm) total and available nutrient concentrations measured in the three landscape positions (topography; B = bottom, S = slope, T = top) and separated by season. Standard errors are shown (n = 20).**

| Site | Season | Topography | C | N | P | $NO_3^- + NH_4^+$ (1M KCl extraction) | Bray P |
|------|--------|-----------|---|---|---|---|---|
| | | | (%) | (%) | (%) | (ppm) | (ppm) |
| Paracou | wet | B | 1.7 +- 0.1 | 0.13 +- 0.01 | 81 +- 4 | 9.6 +- 1.1 | 3.3 +- 0.3 |
| | | S | 2.3 +- 0.2 | 0.16 +- 0.01 | 117 +- 9 | 25.6 +- 3.0 | 1.3 +- 0.1 |
| | | T | 1.9 +- 0.1 | 0.14 +- 0.01 | 76 +- 9 | 15.3 +- 1.7 | 1.3 +- 0.1 |
| | dry | B | 2.5 +- 0.2 | 0.17 +- 0.01 | 89 +- 6 | 4.6 +- 0.4 | 2.8 +- 0.2 |
| | | S | 2.6 +- 0.2 | 0.18 +- 0.01 | 139 +- 10 | 11.6 +- 0.9 | 0.9 +- 0.3 |
| | | T | 2.1 +- 0.1 | 0.14 +- 0.01 | 75 +- 4 | 5.9 +- 0.3 | 1.2 +- 0.1 |
| Nouragues | wet | B | 2.5 +- 0.2 | 0.19 +- 0.02 | 56 +- 5 | 9.3 +- 0.8 | 1.7 +- 0.2 |
| | | S | 2.7 +- 0.1 | 0.20 +- 0.01 | 84 +- 3 | 7.7 +- 0.5 | 0.9 +- 0.1 |
| | | T | 4.2 +- 0.2 | 0.30 +- 0.01 | 311 +- 20 | 16.4 +- 1.0 | 0.9 +- 0.1 |
| | dry | B | 3.1 +- 0.2 | 0.22 +- 0.01 | 67 +- 5 | 9.0 +- 0.9 | 2.1 +- 0.1 |
| | | S | 3.5 +- 0.2 | 0.24 +- 0.01 | 91 +- 4 | 11.2 +- 1.0 | 1.2 +- 0.1 |
| | | T | 4.5 +- 0.2 | 0.30 +- 0.01 | 308 +- 21 | 15.0 +- 0.8 | 1.3 +- 0.1 |

We also added figure 6 comparing our $V_{cmax}$ values to other tropical $V_{cmax}$ values and to the values used in many terrestrial biosphere models.

[Figure]

**Figure 1** *Values of Vcmax at 25°C for sunlit, upper canopy leaves measured in this study compared to (A) different model estimates and (B) other lowland tropical sites growing on nutrient-poor oxisols. The values for the models were adopted from Rogers (2014) for the plant functional types "broadleaf evergreen tropical tree" (Orchidee, O-CN, CLM, AVIM), "tropical tree (oxisol)" (Bethy), "rainforest" (Hybrid), "evergreen broadleaf tree" (Biome-BGC, CTEM), and "broadleaf tree" (Jules). We did not include IBIS, which was also mentioned in Rogers (2014), as the source of the data is unclear. Error bars in (A) show the standard deviation and in (B) the standard error.*

- Some additional data or discussion should be provided in this manuscript. As I know, there were some previous studies for researching the impacts of nutrient additions on forest biophysics, ecophysiology and nutrient cycles in the neighboring Amazon rainforests. A comparison or synthesis among these studies can help cross-validate or falsify the data and results in this study.

The referee suggests to compare on different levels including forest biophysics, ecophysiology and nutrient cycling. In our manuscript, we provide photosynthesis and soil measurements of lowland tropical forests in French Guiana and how these forests respond to nutrient addition at soil and leaf level. Adding a section on which other South American studies provide data on photosynthesis, soil measurements and

compare to other nutrient addition experiments would go far beyond the aim of this data paper. We consider such extensive review outside the scope of this study. Nonetheless, we added a section '5. $V_{cmax}$ of sunlit, upper canopy leaves' to our revised manuscript to put the data into context for the readers:

**"5 $V_{cmax}$ of sunlit, upper canopy leaves**

*TBMs use plant functional types (PFT) to represent broad groupings of plant species that share similar characteristics (e.g. growth form) and roles (e.g. photosynthetic pathway) in ecosystem function (Rogers et al., 2017). Although all TBMs share this approach, they differ from each other in how narrow or broad the PFTs are defined. Depending on the TBM, tropical rainforests belong to "broadleaf evergreen tropical tree", "tropical tree", "rainforest", "evergreen broadleaf tree" or "broadleaf tree" (Figure 2A). Although our mean values of photosynthetic capacity of sunlit, upper canopy leaves are in line with those of other tropical rainforest sites (Figure 2B), many TBMs use estimates for $V_{cmax}$ that are much higher than the estimates from leaf-level measurements (Figure 2A). Only three TBMs (Orchidee, O-CN, and Bethy), which have adopted detailed PFTs, are within the range of our measurements. Some TBMs might obtain more accurate estimates of the global C budget by dividing the adopted PFTs into more detailed PFTs. Hybrid, for example, uses the classification "rainforest", which includes both temperate and tropical rainforest, leading to much higher $V_{cmax}$ values. This is also true for the PFT "evergreen broadleaf tree" used in CTEM and BIOME-BGC, and the PFT "broadleaf tree" used in "JULES".*

[Figure]

**Figure 2** *Values of V$_{cmax}$ at 25°C for sunlit, upper canopy leaves measured in this study compared to (A) different model estimates and (B) other lowland tropical sites growing on nutrient-poor oxisols. The values for the models were adopted from Rogers (2014) for the plant functional types "broadleaf evergreen tropical tree" (Orchidee, O-CN, CLM, AVIM), "tropical tree (oxisol)" (Bethy), "rainforest" (Hybrid), "evergreen broadleaf tree" (Biome-BGC, CTEM), and "broadleaf tree" (Jules). We did not include IBIS, which was also mentioned in Rogers (2014), as the source of the data is unclear. Error bars in (A) show the standard deviation and in (B) the standard error."*